# Put–Call Ratio Volume vs. Open Interest in Predicting Market Return: A Frequency Domain Rolling Causality Analysis

**Sangram Keshari Jena [1],\*, Aviral Kumar Tiwari [2] and Amarnath Mitra [3]**

[1] Department of Finance, IBS Hyderabad, IFHE University, Hyderabad 501203, India
[2] Department of Finance, Control and Law, Montepellier Business School, 34000 Montepelllier, France; aviral.eco@gmail.com
[3] Department of Operations and Quantitative Methods, International Management Institute, New Delhi 110016, India; amarnath.mitra@gmail.com
\* Correspondence: drsangramkjena@gmail.com; Tel.: +91-8826568572

**Abstract:** This study examined the efficacy of the Put–Call Ratio (PCR), a widely used information ratio measured in terms of volume and open interest, in predicting market return at different time scale. Volume PCR was found to be an efficient predictor of the market return in a short period of 2.5 days and open interest PCR in a long period of 12 days. Thus, traders and portfolio managers should use the appropriate PCR depending upon the time horizon of their trade and investment. The results are robust even after controlling for the information generated from the futures market.

**Keywords:** Put–Call Ratio; volume; open interest; frequency-domain roiling causality

## 1. Introduction

Options are a conduit of carrying information into the market, which subsequently leads to stock price changes Grossman (1988). Because informed traders prefer to trade in options market for leverage and low transaction cost Black (1975) and Easley et al. (1998)[1], trading activities of options market measured in terms of volume and open interest are informative to predict the future price of their respective underlying assets. Both options volume and open interest have been used in addition to other factors in modeling early warning system for market crisis Li et al. (2015). Further, as per Jena and Dash (2014), trading volume and open interest represent the strength and potential of price change of the underlying asset, respectively. In addition, traders and technical analysts use open interest data to study the behavior of the underlying asset and design appropriate options strategies. Fodor et al. (2011) found individual call and put open interest have the power to predict future stock return. Most often PCR remains in the news as one of the important and parsimonious information variables used by traders to predict the market return[2]. This ratio is a contrarian indicator of the market by looking at build up options. That means, if there is excessive fall or rise in the market, PCR will move

---

[2] https://economictimes.indiatimes.com/markets/stocks/news/rising-nifty-put-call-ratio-bringssolace-for-bulls-no-big-fall-likely/articleshow/60456238.cmshttps://economictimes.indiatimes.com/markets/stocks/news/spike-in-put-call-ratio-shows-niftymay-correct-1-or-more-in-a-single-session/articleshow/59484386.cmshttps://economictimes.indiatimes.com/markets/stocks/news/rising-put-call-ratio-falling-volatilitysupporting-the-bulls/articleshow/60727912.cms

towards an extreme value based on which the traders can take a contrarian call. Thus, the direction of the market can be determined from the options market by using this most popular indicator, i.e., PCR, which is estimated as follows on a given day for both the measures of trading activity such trading volume and open interest.

PCR (OI) = open interest of put options on a given day/open interest of call options on the same given day

PCR (VOL) = volume of put options on a given day/volume of call options on the same given day

The objective of our study was to discern the efficacy of PCR (OI) and PCR VOL) in predicting the market return. However, is the predictability power of PCR stable across different time scales? Therefore, to answer this question, we investigated the strength and direction of causality at different frequencies using the novel frequency domain causality methodology of Breitung and Candelon (2006) in a rolling framework.

However, few academic studies are found in the literature related to this ratio. Billingsley and Chance (1988) found volume PCR as an effective forecasting tool in predicting the direction of the market. Blau and Brough (2015) in the US market found that current daily PCR of stock options is negatively related to next day's return, thus, as a contrarian trading strategy, PCR has the power of return predictability. Pan and Poteshman (2006) stated that the PCR constructed from buyer initiated volume (signed volume) contains information about future stock prices. Economically, stocks with low PCR are outperforming their higher counterpart stocks by 40 basis points and 1% on the next day and one week, respectively. However, this relative predictability of the PCR is short-lived Pan and Poteshman (2006). Therefore, in our study, we investigated whether the predictability of this ratio is consistent at a different frequencies over a period of time. Unlike Pan and Poteshman (2006), Blau et al. (2014) used unsigned trading volume in their study and investigated the relative information content of PCR and Option to Stock (O/S) ratio. They found that the nature of the information content of PCRs is fleeting at different frequencies. In our study, we tested this fleeting property of PCR at different frequencies in a time-varying framework.

Although information content of option ratios was studied by Roll et al. (2010) and Johnson and So (2012), they both used Option to Stock (O/S) volume ratio[3]. Further, in the literature, only PCR based on volume is studied, ignoring open interest, which is an important trading activity variable. Thus, in our study, in addition to volume PCR, we studied the efficacy of PCR open interest ratio in predicting the future market return. Thus far, existing literature provides one-shot statistic in the time domain in predicting the market return, thereby ignoring the causality dynamics at different frequencies. Thus, we applied Breitung and Candelon's (2006) frequency domain causality for this comparative study of predictability of PCR in both the short and long run. Since in sample frequency domain causality is not robust to structural changes[4] Batten et al. (2017) and Bouri et al. (2017), we estimated out of sample rolling frequency domain causality using a fixed window size of 250 days of observations.

Our contribution to the literature of the derivative market in general and options market, in particular, is threefold.

First, horizon heterogeneity requires information regarding the market at different time periods for trading and investment at different time horizon. We investigated the predictability of option ratios at different frequencies, thereby providing a robust measure for trading and investment at different time horizon for the investors. Second, in addition to volume PCR, we took PCR based on open interest, it being one of the important measures of investors' activity in the derivative market that is currently missing in the literature. Finally, we studied the robustness of our results at the different time periods as well as in the presence of the futures market.

---

3   Other studies on markets include those by Roll et al. (2009) and Chang et al. (2009).
4   We estimated the Bai and Perron (2003) test and the results show five breakpoints in both the cases, i.e., volume PCR and market return, and open interest PCR and market return. The results are available on request.

We found that open interest PCR is an efficient predictor of market return in the long period of 12 days and volume PCR in the short period of 2.5 days. The results are robust after controlling for the information generated in the futures market.

The rest this paper is outlined as follows. Section 2 describes the data and methodology used in the study. The empirical results are presented and discussed in Section 3. Section 4 concludes the paper.

## 2. Data and Methodology

Daily volume and open interest data were collected for the Nifty Index[5] call and put option from the official website of National Stock Exchange of India (NSE)[6] from 1 June 2001 to 16 May 2013. The daily volume and open interest were aggregated across expiry and moneyness for both call and put options and taken for further calculation of daily PCR, the information variable for our study, by following Blau et al. (2014) and Bandopadhyaya and Jones (2008). Put–Call volume (open interest) ratio is the total volume (open interest) of put divided by total volume of call for the day. Log ($P_t/P_{t-1}$) was taken as market return, where $P_t$ and $P_{t-1}$ are closing price of the Nifty index at $t$ and $t-1$, respectively. To control for the information originating from futures trading, we took the trading volume of the NIFTY index futures as a control variable. The descriptive statistics of the variables are presented in Table 1.

**Table 1.** Descriptive statistics of volume put–call ratio (PCRTO), open interest put–call ratio (PCROI), market return (RET) and log Nifty index futures volume (LFTO).

|  | PCROI | PCRTO | RET | LFTO |
|---|---|---|---|---|
| Mean | 1.124 | 0.904 | 0.001 | 12.790 |
| Median | 1.140 | 0.911 | 0.001 | 13.473 |
| Maximum | 3.049 | 2.773 | 0.162 | 14.944 |
| Minimum | 0.210 | 0.136 | −0.163 | 6.862 |
| Std. Dev. | 0.414 | 0.264 | 0.019 | 1.703 |
| Skewness | 0.155 | 0.370 | −0.136 | −1.420 |
| Kurtosis | 3.177 | 4.969 | 12.447 | 3.840 |
| Jarque–Bera | 11.491 | 399.741 | 8065.200 | 791.801 |
| Probability | 0.003 *** | 0.000 *** | 0.000 *** | 0.000 *** |
| Observations | 2168 | 2168 | 2167 | 2168 |
| Augmented Dickey–Fuller test statistic (*p*-values) | −7.892 (0.000 ***) | −7.682 (0.000 ***) | −46.776 (0.000 ***) | −3.314 (0.014 **) |

*** and ** represent significance at 1% and 5% levels, respectively.

An average PCROI (PCRTO) greater than one (less than one) indicates a positive (negative) market sentiment. This justifies the PCROI taken in this study in addition to PCRTO. All the series were stationary. Moreover, since all the series were non-normal and fat-tailed, it further justified our methodology.

For the purpose of estimation, we used the frequency domain Granger causality (GC) test by following Bouri et al. (2017) as the widely utilized GC test (Granger 1969) is the one-shot measure of GC, which is assumed to be constant over time and frequency. Hosoya (1991) suggested that the causal influence may change across frequencies; nonetheless, they pointed out estimation difficulties owing to nonlinearities of the data to measure GC, which was made possible by Breitung and Candelon (2006)[7]

---

[5] The bellwether index of National Stock Exchange of India (NSE) represents 65% of the total market capitalization and 12 sectors of the economy.
[6] www.nseindia.com.
[7] Yamada and Yanfeng (2014) through theoretical evaluation tested the usefulness of the methodology even at a frequency close to zero.

by imposing linear restrictions on the autoregressive parameters in a VAR model and thus allowing for the estimation of the frequency domain approach to causality at different frequency bands. Several studies have used this approach (for example, Tiwari et al. 2014, 2015 and references therein), therefore we provide a small introduction to the approach.

Let us present an equation of a stationary VAR framework of two series $x_t$ and $y_t$ as follows:

$$x_t = a_1 x_{t-1} + \ldots + a_p x_{t-p} + \beta_1 y_{t-1} + \ldots + \beta_p y_{t-p} + \varepsilon_t \tag{1}$$

The null hypothesis that $y_t$ does not Granger-cause $x_t$ at frequency ($\omega$) in Equation (1) is tested by,

$$H_0 : R(\omega)\beta = 0 \tag{2}$$

where $\beta$ is the vector of the coefficients of $y_t$ i.e., $\beta = [\beta_1, \beta_1, \ldots \beta_p]$ and

$$R(\omega) = \begin{bmatrix} \cos(\omega) \cos(2\omega) \ldots \cos(p\omega) \\ \sin(\omega) \sin(2\omega) \ldots \sin(p\omega) \end{bmatrix} \tag{3}$$

According to the Breitung and Candelon (2006), an ordinary *F* statistic for Equation (2) can be used to test the hull hypothesis at any frequency interval (i.e., $\omega \in (0, \pi)$) as it is approximately distributed as $F(2, T - 2p)$. Further, for the purpose of interpretations in time framework, the frequency parameter $\omega$ (omega) can be used to obtain the time period of the causality in days (T) by using formula $T = 2\pi/\omega$.

## 3. Empirical Analysis

First, we estimated the VAR granger causality[8] (both unconditional and conditional on index futures volume) for the purpose of comparisons with the results of causality estimated at the frequency domain. The results are presented in Table 2.

**Table 2.** VAR Granger causality.

|  | Unconditional Chi-sq. Test Statistic (*p*-Values) | Conditional Chi-sq. Test Statistic (*p*-Values) |
|---|---|---|
| PCR TO $\neq$> RET | 3.548 (0.470) | 5.887 (0.207) |
| NIFTY RET $\neq$> PCR TO | 23.403 (0.000 ***) | 26.213 (0.000 ***) |
| PCR OI $\neq$> RET | 9.326 (0.009 ***) | 15.999 (0.000 ***) |
| NIFTY RET $\neq$> PCR OI | 27.469 (0.000 ***) | 36.878 (0.000 ***) |

*** indicates significance at 1% level. $\neq$> refers to "does not granger cause".

No causality was observed from PCRTO to market return. PCROI Granger causes market return. However, this one-shot measure of GC may not hold across frequencies owing to nonlinearities of the data Hosoya (1991). This further justifies the application of Breitung and Candelon's (2006) methodology and the results are discussed in the following section.

Figure 1 presents the frequency domain causality from put–call ratio volume (PCRTO) and open interest (PCROI) to market return[9]. The blue solid line shows the Granger causality from PCRTO to market return, which is insignificant throughout at both 5% and 10% levels. That means volume PCR does not have predictive power of market return, which is against the popular belief of being a sentiment indicator Open interest put–call ratio (PCROI) significantly (at 5% level) Granger causes

---

8   We are thankful to the anonymous referees for this suggestion.
9   The descriptive statistics of the F-statics of the frequency domain causality results are presented in Appendix A Table A1.

market return in long run at a frequency band 0.51 corresponding to 12 days and above. At the 10% level of significance, it leads the market return between a frequency bands of 0.93–0.51 corresponding to 6–12 days. It implies that open interest PCR is a better predictor of market return than its volume counterpart in the long run. None of these ratios can predict in the short run.

Figures 2 and 3 present the rolling frequency domain causality from PCR volume and open interest to market return. Notably, short term causalities were estimated at frequency of 2.5 corresponding to 2–3 days, as presented in Figure 3, and long-term causality at frequency of 0.50 corresponding to 12–13 days is estimated, aas presented in Figure 2.

The long-term rolling causality in Figure 2 is consistent with the results reported in the in-sample analysis. The predictability of open interest PCR dominates its volume counterparts, as indicated by the dominant and significant peak of its frequency curve from June 2003 to June 2006 and from December 2010 to February 2012. One thing that stands out is that, during the 2008 financial crisis and after the 2012 European sovereign debt crisis, none of the indicators is significant in predicting the market return. Thus, the traders should carefully use these ratios during market crisis.

However, over a short period of 2.5 days, the reported rolling causality in Figure 3 volume PCR dominates its open interest counterpart, which is in stark contrast to the in-sample result. Another interesting thing that stands out from the figures is that, in the short term, volume PCR is a good predictor of market return. Moreover, it is a good predictor during the 2008 financial crisis, as evident from the higher amplitude volume PCR frequency curve, which is significant at 5% level. However, open interest PCR in the short run dominates for a brief period from April 2011 to November 2011[10]. Our results supplement the findings of Pan and Poteshman (2006) and Blau et al. (2014) that the volume PCR is short lived and fleeting, respectively.

---

[10] We also estimated conditional frequency domain rolling causality analysis after controlling for the futures market activities. The results are quite similar and available upon request.

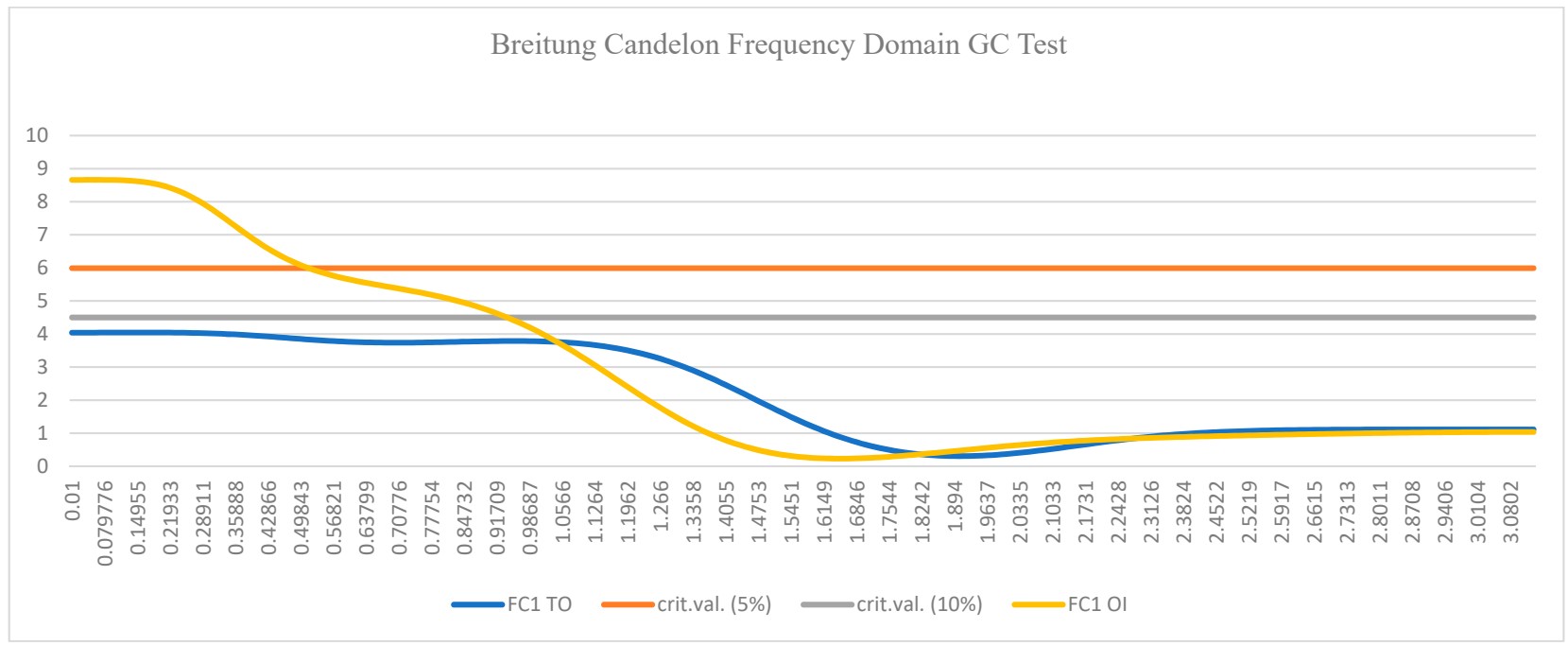

**Figure 1.** Full sample period frequency domain causality from volume put–call ratio (PCRTO) to market return is represented by blue solid line (FC1 TO) and the yellow solid line (FC1 OI) shows from open interest put–call ratio (PCROI) to market return. The frequencies (omega, ω) are on x-axis, and F-statistics testing the null hypothesis of no Granger causality are on y-axis. The horizontal red solid line and grey solid line indicate the 5% and 10% critical values, respectively.

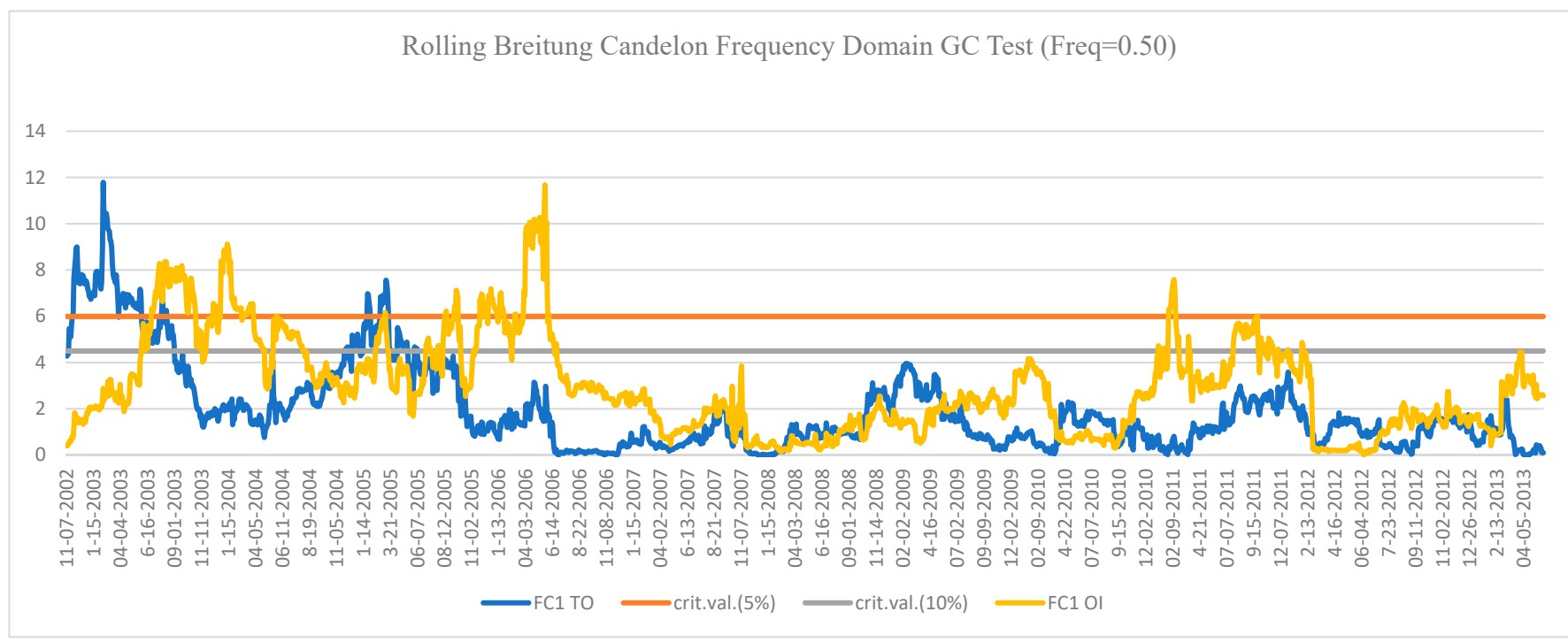

**Figure 2.** Long-run ($\omega = 0.50$ or 12 days) rolling window frequency domain causality. The x-axis represents the date and y-axis shows the F-statistics testing the null hypothesis of no Granger causality from volume Put–call ratio (FC1 TO) and open interest Put–call ratio (FC1 OI) to market return. The horizontal red solid line and grey solid line indicate the 5% and 10% critical values, respectively. The blue solid line (FC1 TO) and yellow solid line (FC1 OI) show long run causality from volume PCR and open interest put ratio, respectively, to market return.

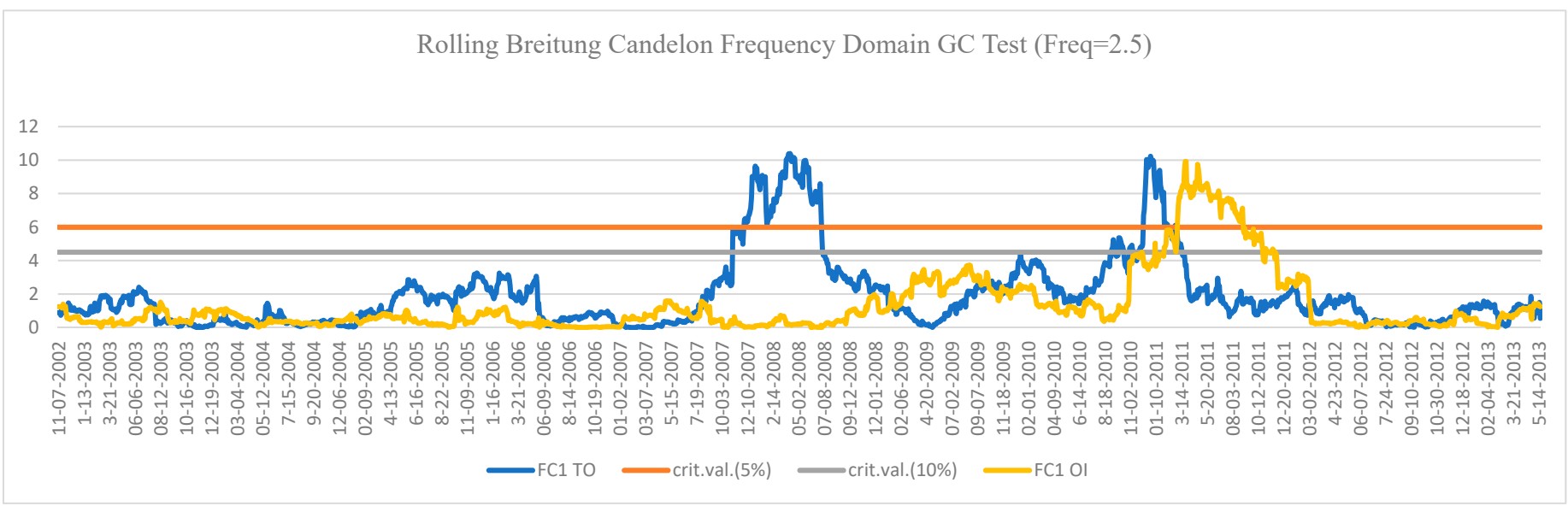

**Figure 3.** Short-run ($\omega$ = 2.5 or 2.5 days) rolling window frequency domain causality. The x-axis represents the date and y-axis shows the F-statistics testing the null hypothesis of no Granger causality from volume Put–call ratio (FC1 TO) and open interest Put–call ratio (FC1 OI) to market return. The horizontal red solid line and grey solid line indicate the 5% and 10% critical values, respectively. The blue solid line (FC1 TO) and yellow solid line (FC1 OI) show short run causality from volume PCR and open interest put ratio, respectively, to market return.

## 4. Conclusions

Extending the prior research relating to informational role of derivative market in general and option market in particular, this study examined the informational efficiency of volume and open interest PCR in predicting the market return and its implication for traders and portfolio managers. First, we studied the efficiency of the PCR at different frequencies and the results were tested in an out of sample forecasting exercises in a rolling frequency domain causality framework. The Granger causality from PCR to market return varies across the frequencies. Long-run causality was observed from open interest PCR to market return corresponding to time period of 12 days. In the short run, corresponding to 2.5 days, volume PCR Granger causes market return. Thus, traders and portfolio managers should use the appropriate PCR at the different time period in predicting a market return for trading and investment. In addition, unlike in the long run, the short-run volume PCR holds the predictability of market return during crisis period. Further, our findings are robust even after controlling for the information generated from futures market. In the future, this study could be extended to effectiveness of PCR ratios across maturity and moneyness of the index options as well as stock options.

**Author Contributions:** All authors contributed equally to this work.

**Acknowledgments:** This is an original research work. We are thankful to the anonymous reviewers for their constructive comments which have contributed to the improvement in the quality of the manuscript.

**Conflicts of Interest:** The authors declare no conflict of interest.

## Appendix A

**Table A1.** Descriptive statistics of the F-statics of the frequency domain causality.

|  | In Sample | | Out of Sample (Freq. 0.5) | | Out of Sample (Freq. 2.5) | |
|---|---|---|---|---|---|---|
|  | FC1 TO | FC1 OI | FC1 TO | FC1 OI | FC1 TO | FC1 OI |
| Mean | 2.195 | 2.785 | 1.953 | 2.955 | 1.942 | 1.323 |
| Median | 1.332 | 1.026 | 1.359 | 2.571 | 1.350 | 0.565 |
| Standard Deviation | 1.441 | 2.772 | 1.888 | 2.112 | 2.157 | 1.849 |
| Kurtosis | −1.783 | −0.620 | 3.032 | 0.621 | 4.114 | 5.483 |
| Skewness | 0.140 | 0.929 | 1.719 | 0.948 | 2.050 | 2.380 |
| Minimum | 0.308 | 0.232 | 0.002 | 0.009 | 0.000 | 0.001 |
| Maximum | 4.046 | 8.662 | 11.784 | 11.674 | 10.391 | 9.919 |
| No of Obs. | 314 | 314 | 1917 | 1917 | 1917 | 1917 |

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
