# Peer review of "Put–Call Ratio Volume vs. Open Interest in Predicting Market Return: A Frequency Domain Rolling Causality Analysis"

_economies, doi:10.3390/economies7010024_

Round 1

Reviewer 1 Report

The introduction section should explain how this paper differs from previous studies. 

The contribution of this paper needs to be explicitly stated in the introduction section.

There should be a presentation or discussion of the empirical model to be estimated. This should be presented before section 2 which deals with methodology and data.

The conclusion section should also have few sentences of the contribution of this paper.

Author Response

Reviewer - 1

Comments and Suggestions for Authors

The introduction section should explain how this paper differs from previous studies. 

Reply: We are thankful for such valuable comments.

We have studied the validity of the two claims made in the previous literature with regard to Put Call Ratio in paragraph 3 of the introduction on page no 2. The paragraph is as follows.

Most often Put Call Ratio (PCR) remains in news talked about and used by practitioners as one of the important and parsimonious information variables in predicting the market return[1]. However, few academic studies are found related to this ratio. Pan and Potashman (2006) state that the put-call ratio constructed from buyer initiated volume (signed volume) contains information about future stock prices. Economically, stocks with the low put-call ratio are outperforming their higher counterpart stocks by 40 basis points and 1% in the next day and one week respectively. However, this relative predictability of the put-call ratio is short-lived, Pan and Poteshman (2006). Therefore in our study, we investigate whether the predictability of this ratio is consistent at a different frequency over a period of time. Unlike Pan and Poteshman (2006), Balu et.al (2014) used unsigned trading volume in their study and investigate the relative information content of PCR and Option to Stock (O/S) ratio. They find that the nature of the information content of put-call ratios is fleeting at a different frequency. In our study, we test this fleeting property of PCR at a different frequency in a time-varying framework.

The contribution of this paper needs to be explicitly stated in the introduction section.

Reply: As suggested we have presented the contribution of our paper in paragraph 2, page no 3 as follows.

Our contributions to the literature of the derivative market in general and options market, in particular, are three folds.

First, horizon heterogeneity requires information regarding the market at different time periods for trading and investment at different time horizon. We have investigated the predictability of option ratios at different frequency thereby providing a robust measure for trading and investment at different time horizon for the investors. Second, in addition to volume put-call ratio, we have taken put-call ratio based on open interest being one of the important measures of investors' activity in the derivative market, which is so far missing in the literature. Finally, we have studied the robustness of our results at the different time period and also in the presence of the futures market.

There should be a presentation or discussion of the empirical model to be estimated. This should be presented before section 2 which deals with methodology and data.

Reply: As suggested we have presented and discussed the methodology in section 2 starting with the last paragraph of the page no 3 and continued up to mid of the page no 4   

The conclusion section should also have a few sentences of the contribution of this paper.

Reply: As suggested we have mentioned the contribution of our paper to the traders and investors as follows.

Thus traders and portfolio managers should use the appropriate PCR at the different time period in predicting a market return for trading and investment.  

[1] https://economictimes.indiatimes.com/markets/stocks/news/rising-nifty-put-call-ratio-bringssolace-for-bulls-no-big-fall-likely/articleshow/1511405950pxs

https://economictimes.indiatimes.com/markets/stocks/news/spike-in-put-call-ratio-shows-nifty may-correct-1-or-more-in-a-single-session/articleshow/1487109650pxs

https://economictimes.indiatimes.com/markets/stocks/news/rising-put-call-ratio-falling-volatility supporting-the-bulls/articleshow/1518197800pxs

Reviewer 2 Report

This is clearly a premature submission. There is extensive need for copy editing. Grammatical and punctuation errors are everywhere.

The paper comes across as an author hunting for an application of a methodology, quickly running some analyses and throwing it into a short paper. It does not come across as particularly useful nor does the author seem to demonstrate a strong background in finance.

The literature on the methodological side is sparse. While the methodology is interesting in itself and the application to options markets is potentially interesting, the haphazard execution is not nearly a strong enough attempt to warrant peer reviewed publication in any reasonable academic journal.

The author should thouroguly copy edit and build out the paper more. I suggest deeper literature review. 

You could start researching more on this topic with the following paper and those cited therein. 

Hiroshi Yamada & Wei Yanfeng (2014) Some Theoretical and Simulation Results on the Frequency Domain Causality Test, Econometric Reviews, 33:8, 936-947, DOI: 10.1080/07474938.2013.808488

You perform frequency domain causality testing and make strong claims about forecastability. How about actually trying to make some forecasting models based on the evidence you suggest to be meaningful? Then presenting in- and out-of-sample  is a must. Further, you should demonstrate how profitable such a strategy would be.

Author Response

Reviewer 2

Comments and Suggestions for Authors

This is clearly a premature submission. There is extensive need for copy editing. Grammatical and punctuation errors are everywhere.

Reply: Thanks for the valuable comments. We have copy edited and taken care of the Grammatical and punctuation errors are everywhere.

The paper comes across as an author hunting for an application of a methodology, quickly running some analyses and throwing it into a short paper. It does not come across as particularly useful nor does the author seem to demonstrate a strong background in finance.

Reply: We have disused the significance of the issues and the motivation to apply this methodology to address the issues in paragraph 3 of the introduction on page no 2. The paragraph is as follows.

Most often Put Call Ratio (PCR) remains in news talked about and used by practitioners as one of the important and parsimonious information variables in predicting the market return[1]. However, few academic studies are found related to this ratio. Pan and Potashman (2006) state that the put-call ratio constructed from buyer initiated volume (signed volume) contains information about future stock prices. Economically, stocks with the low put-call ratio are outperforming their higher counterpart stocks by 40 basis points and 1% in the next day and one week respectively. However, this relative predictability of the put-call ratio is short-lived, Pan and Poteshman (2006). Therefore in our study, we investigate whether the predictability of this ratio is consistent at a different frequency over a period of time. Unlike Pan and Poteshman (2006), Balu et.al (2014) used unsigned trading volume in their study and investigate the relative information content of PCR and Option to Stock (O/S) ratio. They find that the nature of the information content of put-call ratios is fleeting at a different frequency. In our study, we test this fleeting property of PCR at a different frequency in a time-varying framework.

The literature on the methodological side is sparse. While the methodology is interesting in itself and the application to options markets is potentially interesting, the haphazard execution is not nearly a strong enough attempt to warrant peer reviewed publication in any reasonable academic journal.

Reply: Now as suggested the paper is properly structured and the contributions of this paper are presented in paragraph 2 , page no 3 as follows.

Our contributions to the literature of the derivative market in general and options market, in particular, are three folds.

First, horizon heterogeneity requires information regarding the market at different time periods for trading and investment at different time horizon. We have investigated the predictability of option ratios at different frequency thereby providing a robust measure for trading and investment at different time horizon for the investors. Second, in addition to volume put-call ratio, we have taken put-call ratio based on open interest being one of the important measures of investors' activity in the derivative market, which is so far missing in the literature. Finally, we have studied the robustness of our results at the different time period and also in the presence of the futures market.

The author should thoroughly copy edit and build out the paper more. I suggest deeper literature review. 

Reply: As suggested it is thoroughly copy edited. Further, as you rightly said in the previous comments the literature is sparse. However, we have presented few literature relevant to our study in page no 2 as follows.

Most often Put Call Ratio (PCR) remains in news talked about and used by practitioners as one of the important and parsimonious information variables in predicting the market return[2]. However, few academic studies are found related to this ratio. Pan and Potashman (2006) state that the put-call ratio constructed from buyer initiated volume (signed volume) contains information about future stock prices. Economically, stocks with the low put-call ratio are outperforming their higher counterpart stocks by 40 basis points and 1% in the next day and one week respectively. However, this relative predictability of the put-call ratio is short-lived, Pan and Poteshman (2006). Therefore in our study, we investigate whether the predictability of this ratio is consistent at a different frequency over a period of time. Unlike Pan and Poteshman (2006), Balu et.al (2014) used unsigned trading volume in their study and investigate the relative information content of PCR and Option to Stock (O/S) ratio. They find that the nature of the information content of put-call ratios is fleeting at a different frequency. In our study, we test this fleeting property of PCR at a different frequency in a time-varying framework.

However, although information content of option ratios are studied by Roll et.al. (2010) and Johnson and So (2010), in both the studies they have used Option to Stock (O/S) volume ratio[3]. Further, it is found in the literature that only PCR based on volume is studied ignoring the open interest being an important trading activity variable. So in our study in addition to volume PCR, we have studied the efficacy of PCR open interest ratio in predicting the future market return.   Moreover, so far existing literature provides one-shot statistic in the time domain in predicting the market return thereby ignoring the causality dynamics at a different frequency. So we have applied Breitung and Candelon (2006) frequency domain causality for this comparative study of predictability of PCR at both short-and-long -run.  Since in sample frequency domain causality is not robust to structural changes[4] Batten et.al. (2017)  and Bouri et.al. (2017), we have estimated out of sample rolling frequency domain causality using a fixed window size of 250 days observations. 

You could start researching more on this topic with the following paper and those cited therein. 

Hiroshi Yamada & Wei Yanfeng (2014) Some Theoretical and Simulation Results on the Frequency Domain Causality Test, Econometric Reviews, 33:8, 936-947, DOI: 10.1080/07474938.2013.808488

Reply: As suggested we have gone through the paper and discussed the significance and robustness of the Frequency domain causality through the following footnote in page no 3

Hiroshi Yamada & Wei Yanfeng (2014) through theoretical and evaluation test the usefulness of the methodology even at a frequency close to zero.

You perform frequency domain causality testing and make strong claims about forecastability. How about actually trying to make some forecasting models based on the evidence you suggest to be meaningful? Then presenting in- and out-of-sample  is a must. Further, you should demonstrate how profitable such a strategy would be.

Reply: We are very much thankful to the referees for such as valuable suggestions. However, the objective of our study is to model the predictability of option ratios for different frequencies in the context of stock index. Future research could be extended to out of sample forecasting, implementation of trading based forecasting and evaluation of performance.  

[1] https://economictimes.indiatimes.com/markets/stocks/news/rising-nifty-put-call-ratio-bringssolace-for-bulls-no-big-fall-likely/articleshow/1511405950pxs

https://economictimes.indiatimes.com/markets/stocks/news/spike-in-put-call-ratio-shows-nifty may-correct-1-or-more-in-a-single-session/articleshow/1487109650pxs

https://economictimes.indiatimes.com/markets/stocks/news/rising-put-call-ratio-falling-volatility supporting-the-bulls/articleshow/1518197800pxs

[2] https://economictimes.indiatimes.com/markets/stocks/news/rising-nifty-put-call-ratio-bringssolace-for-bulls-no-big-fall-likely/articleshow/1511405950pxs

https://economictimes.indiatimes.com/markets/stocks/news/spike-in-put-call-ratio-shows-nifty may-correct-1-or-more-in-a-single-session/articleshow/1487109650pxs

https://economictimes.indiatimes.com/markets/stocks/news/rising-put-call-ratio-falling-volatility supporting-the-bulls/articleshow/1518197800pxs

[3] Few other studies on markets are Roll et.al. (2009) and Chang et.al. (2009)

[4] We have estimated Bai-Perron (2003) test and the results show 5 breakpoints in both the cases i.e. volume PCR and market return and open interest PCR and market return. The results are available on request.

Reviewer 3 Report

The paper examines the predictive power of put-call ratio using a frequency domain Granger causality test as in Breitung and Candelon (2006). The issue is interesting and important. But more empirical evidence is needed before the paper is publishable.

(1) The authors use aggregated call and put option open interest and volume data.  The authors should provide a summary of the existing literature (top tier papers) on the US evidence and let readers know if US evidence uses aggregate data or individual call and put data.  If there are some top-tier papers that use individual call and put option data, the authors should follow and provide the empirical evidence from individual call and put option data.

(2) The authors should provide more description of the F-statistic and explanation for the test. Are there any restriction on the properties of the time-series data?  What is the F-statistic like?

(3) The authors should provide a table summarizing the mean, median, standard deviation of the market return and put call open interest and volume data.

(4) The authors should check whether open interest and trading  volume has a trend by using a unit root test and report the test statistics.

(5) The authors should also implement the traditional Granger causality test and compare with the frequency domain Granger causality test.

(6) The authors should prepare  a table summarizing the frequency domain Granger causality tests for some key frequencies. 

(7) The authors should do a search regarding the put-call ratio predicability literature. The authors can focus on top tiers journals over the past 5-10 years.  This will make the paper look much more serious and sophisticated.

Author Response

Responses to Reviewers’ Remarks

Title: Put Call Ratio Volume vs. Open Interest in predicting market return: A frequency domain rolling causality analysis

Submitted to Economies

We would like to thank the Editors of the Economies for giving us the opportunity to revise our work. As suggested, we have carefully addressed all the comments of the reviewers. Our responses, indicated in the yellow color highlight, discuss and explain in detail the responses to the reviewers’ comments. Part or all of those responses has been added to the revised manuscript or new version of the paper.

Reviewer 3

Comments and Suggestions for Authors

The paper examines the predictive power of put-call ratio using a frequency domain Granger causality test as in Breitung and Candelon (2006). The issue is interesting and important. But more empirical evidence is needed before the paper is publishable.

Reply: many thanks dear referee for being positive about the manuscript and making your comments and suggestion which we have considered with great care.

(1) The authors use aggregated call and put option open interest and volume data.  The authors should provide a summary of the existing literature (top tier papers) on the US evidence and let readers know if US evidence uses aggregate data or individual call and put data.  If there are some top-tier papers that use individual call and put option data, the authors should follow and provide the empirical evidence from individual call and put option data.

Reply: We are thankful for this comment. We have followed (US evidences)  Blau & Brough (2015) Balu et.al (2014) and Bandopadhyaya and Jones (2008)  and taken daily aggregate volume and open interest for estimating put call ratio.

Further, as suggested we have mentioned in section 2 ( Data and Methodology) that by following Blau & Brough (2015),  Balu et.al (2014) and Bandopadhyaya and Jones (2008), we have estimated aggregated put call ratio.

(2) The authors should provide more description of the F-statistic and explanation for the test. Are there any restriction on the properties of the time-series data?  What is the F-statistic like?

Reply: Thanks for the suggestion, It is explained in the last paragraph of methodology in section 2

According to the Breitung and Candelon (2006) an ordinary  statistic for Eq.2 can be used to test the hull hypothesis at any frequency interval (i.e., ) as it is approximately distributed as . Further, for the purpose of interpretations in time framework the frequency parameter ω (omega) can be used to obtain the time period of the causality in days (T) by using formula T= 2π/ω. The descriptive stat. of F-statics is provided in the appendix (Table A-1) as follows.

In Sample

Out of Sample (Freq. 0.5)

Out of Sample (Freq. 2.5)

FC1 TO

FC1 OI

FC1 TO

FC1 OI

FC1 TO

FC1 OI

Mean

2.195

2.785

1.953

2.955

1.942

1.323

Median

1.332

1.026

1.359

2.571

1.350

0.565

Standard Deviation

1.441

2.772

1.888

2.112

2.157

1.849

Kurtosis

-1.783

-0.620

3.032

0.621

4.114

5.483

Skewness

0.140

0.929

1.719

0.948

2.050

2.380

Minimum

0.308

0.232

0.002

0.009

0.000

0.001

Maximum

4.046

8.662

11.784

11.674

10.391

9.919

No of Obs.

314

314

1917

1917

1917

1917

(3) The authors should provide a table summarizing the mean, median, standard deviation of the market return and put call open interest and volume data.

Reply: Thanks for the suggestion.  The descriptive statistics are presented through Table 1

(4) The authors should check whether open interest and trading volume has a trend by using a unit root test and report the test statistics.

Reply: Thanks for the suggestion

Stationarity is tested using ADF test. The results are reported in the Table 1.

(5) The authors should also implement the traditional Granger causality test and compare with the frequency domain Granger causality test. 

Reply:  Thanks for the suggestions. We have estimated VAR granger causality and results are reported in Table 2

(6) The authors should prepare  a table summarizing the frequency domain Granger causality tests for some key frequencies.  

Reply: we are thankful for this comment. The descriptive stat. of F-statics is provided in the appendix (Table A-1) as follows.

In Sample

Out of Sample (Freq. 0.5)

Out of Sample (Freq. 2.5)

FC1 TO

FC1 OI

FC1 TO

FC1 OI

FC1 TO

FC1 OI

Mean

2.195

2.785

1.953

2.955

1.942

1.323

Median

1.332

1.026

1.359

2.571

1.350

0.565

Standard Deviation

1.441

2.772

1.888

2.112

2.157

1.849

Kurtosis

-1.783

-0.620

3.032

0.621

4.114

5.483

Skewness

0.140

0.929

1.719

0.948

2.050

2.380

Minimum

0.308

0.232

0.002

0.009

0.000

0.001

Maximum

4.046

8.662

11.784

11.674

10.391

9.919

No of Obs.

314

314

1917

1917

1917

1917

(7) The authors should do a search regarding the put-call ratio predicability literature. The authors can focus on top tiers journals over the past 5-10 years.  This will make the paper look much more serious and sophisticated.

Reply: We are thankful to the referee for this suggestion. Although literatures are few on put –call ratio, we have discussed following additional two papers on US market in the literature.

Blau, B. M., & Brough, T. J. (2015). Are put-call ratios a substitute for short sales?. Review of Derivatives Research18(1), 51-73.

Billingsley, R. S., & Chance, D. M. (1988). Put-call ratios and market timing effectiveness. Journal of Portfolio Management15(1), 25.

Reviewer 4 Report

Comments on the paper entitled “ Put Call Ratio Volume vs. Open Interset in Prexdicting Mrket Return. A Frequency Domain Roling Causality Analysis”

In this short article the authors examine the validity of the Put Call Ratio in predicting mrket returns. For this purpose, the use a frequency domain causlity test proposed in Breitung and Candelon (2006). Their results indicate that Volume PCR is an efficient predictor of the market retun in the short run and ope interst PCR dominates in the long run.

This is an interesting paper investigating a relevant topic with an updated technique.

I recommend its publication once the authors take into account the comment below.

Recommendation:     Minor / Major revision

Comments:

Footnote 1 in page 1:             All the references included in this footnote are quite old. Please include some more updated ones.

The English is in general quite deficient and should be improved in the revised version.

Line 76:          “ The rest of the work is outlined as follows: …”

Section 2, lines 88 and 89:     Granger (1969) and Hosoya (1991) do not appear in the Reference section.

Line 95:          Tiwari et al. (2014, 2015) are not referenced.

Section 2:        The functional form of the test statistic is Breitung and Candelon (12004) is not provided. Please, explain this method futher and include its analytical expression.

Footnote 2 in Section 2:         Hiroshi Yamada and Wei Yanfeng (2014) is not referenced. Include only surnames.

Line 113:        Replace “ granger “ by “ Granger ”.

Line 131:        Replace “ it’s “ by “ it is “.

Section 4 should be further elaborated, including some more comments on the implications of the results obtained, and research avenues for future work.

References

Breitung, J. nad B. Candelon, 2006, Testing for short and long run causality. A frequency domain approach, Journal of Econometrics 132, 2, 363-378.

Author Response

Responses to Reviewers’ Remarks

Title: Put Call Ratio Volume vs. Open Interest in predicting market return: A frequency domain rolling causality analysis

Submitted to Economies

We would like to thank the Editors of the Economies for giving us the opportunity to revise our work. As suggested, we have carefully addressed all the comments of the reviewers. Our responses, indicated in the yellow color highlight, discuss and explain in detail the responses to the reviewers’ comments. Part or all of those responses has been added to the revised manuscript or new version of the paper.

Reviewer 4

Comments and Suggestions for Authors

Comments on the paper entitled “ Put Call Ratio Volume vs. Open Interest in Predicting Market Return. A Frequency Domain Rolling Causality Analysis”

In this short article the authors examine the validity of the Put Call Ratio in predicting market returns. For this purpose, the use a frequency domain causality test proposed in Breitung and Candelon (2006). Their results indicate that Volume PCR is an efficient predictor of the market retun in the short run and ope interst PCR dominates in the long run.

This is an interesting paper investigating a relevant topic with an updated technique.

I recommend its publication once the authors take into account the comment below.

Recommendation:     Minor / Major revision

Reply: many thanks dear referee for being positive about the manuscript and making your comments and suggestion which we have considered with great care.

Comments:

Footnote 1 in page 1:             All the references included in this footnote are quite old. Please include some more updated ones.

Reply : Thanks for this valuable suggestion. We have incorporated the following updated references to footnote 1

Dubrian et. al. (2018), Ryu (2015), Cao and Ye (2016) and Chordia et.al. (2018)

Dubrian, B., Fung, S., & Lovelandrobert, R. (2018). The Informational Role of Options Markets: Evidence from FOMC Announcements. Journal of Banking & Finance.

Ryu, D. (2015). The information content of trades: An analysis of KOSPI 200 index derivatives. Journal of Futures Markets35(3), 201-221.

Cao, H. H., & Ye, D. (2016). Transaction Risk, Derivative Assets, and Equilibrium. Quarterly Journal of Finance6(01), 1650001.

Chordia, Tarun and Kurov, Alexander and Muravyev, Dmitriy and Subrahmanyam, Avanidhar, Index Option Trading Activity and Market Returns (May 17, 2018). Available at SSRN: http://dx.doi.org/10.2139/ssrn.2798390

The English is in general quite deficient and should be improved in the revised version.

Reply: Thanks for the suggestion. This has been taken care of in the revised manuscript.

Line 76:          “ The rest of the work is outlined as follows: …”

Reply: Thanks for pointing out the mistake. It has been rectified.

Section 2, lines 88 and 89:     Granger (1969) and Hosoya (1991) do not appear in the Reference section.

Reply:  It is referenced now as follows.

Granger, C. W. (1969). Investigating causal relations by econometric models and cross-spectral methods. Econometrica: Journal of the Econometric Society, 424-438.

Hosoya, Y. (1991). The decomposition and measurement of the interdependency between second-order stationary processes. Probability theory and related fields88(4), 429-444.

Line 95:          Tiwari et al. (2014, 2015) are not referenced.

Reply: Thanks for identifying this mistake. We have referenced as follows.

Tiwari, A. K., Mutascu, M. I., Albulescu, C. T., & Kyophilavong, P. (2015). Frequency domain causality analysis of stock market and economic activity in India. International Review of Economics & Finance39, 224-238.

Tiwari, A., Arouri, M., & Teulon, F. (2014). Oil prices and trade balance: a frequency domain analysis for India. Economics Bulletin34(2), 663-680.

Section 2:        The functional form of the test statistic is Breitung and Candelon (12004) is not provided. Please, explain this method futher and include its analytical expression.

Reply: Thanks for the suggestion, It has been explained in the last paragraph of methodology part of  section 2

According to the Breitung and Candelon (2006) an ordinary  statistic for Eq.2 can be used to test the hull hypothesis at any frequency interval (i.e., ) as it is approximately distributed as . Further, for the purpose of interpretations in time framework the frequency parameter ω (omega) can be used to obtain the time period of the causality in days (T) by using formula T= 2π/ω

Footnote 2 in Section 2:         Hiroshi Yamada and Wei Yanfeng (2014) is not referenced. Include only surnames.

Reply: Thanks for pointing out the mistake. Now it is cited in foot note with surname only i.e. Yamada & Yanfeng (2014) and also added to the reference as follows.

Yamada, H., & Yanfeng, W. (2014). Some theoretical and simulation results on the frequency domain causality test. Econometric Reviews33(8), 936-947

Line 113:        Replace “ granger “ by “ Granger ”.

Reply: Thanks for pointing out the mistake. It is rectified.

Line 131:        Replace “ it’s “ by “ it is “.

Reply: Thanks for pointing out the mistake. It is rectified.

Section 4 should be further elaborated, including some more comments on the implications of the results obtained, and research avenues for future work.

Reply: We are thankful for this valuable suggestion. The section 4 part has been elaborated as follows.

Extending the prior research relating to informational role of derivative market in general and option market in particular, the study examined the informational efficiency of volume and open interest PCR in predicting the market return and its implication for the traders and portfolio managers. First time we have studied the efficiency of the PCR at different frequencies and also the results are tested in an out of sample forecasting exercises in a rolling frequency domain causality framework. Firstly, the Granger causality from PCR to market return varies across the frequencies. Secondly, Long run causality is observed from open interest PCR to market return corresponding to time period of 12 days. In short run corresponding a time period of 2.5 days, volume PCR Granger causes market return. Thus traders and portfolio managers should use the appropriate PCR at the different time period in predicting a market return for trading and investment. In addition, unlike in long-run, in short-run volume PCR holds the predictability of market return during crisis period. Further, our findings are robust even after controlling for the information generated from futures market. The future study in this regard could be extended to effectiveness of PCR ratios across maturity and moneyness of the index options and also to stock options. 

References

Breitung, J. nad B. Candelon, 2006, Testing for short and long run causality. A frequency domain approach, Journal of Econometrics 132, 2, 363-378.

Reviewer 5 Report

This paper is analyzing an interesting topic.

The following points should be clarified :

1. Justification on the sample chosen. Recommendation (NIFTY Mid Cap, NIFTY Small Cap, may be analyzed. if author/s find common result, so the finding will be more valuable and generalizable).

2. Literature review section should be improved and the gap should be more clearly addressed.

3. Plagiarism rate is quite high (20%). 

4. Conclusion should be enhanced, specially the part related to the findings.

Author Response

Responses to Reviewers’ Remarks

Title: Put Call Ratio Volume vs. Open Interest in predicting market return: A frequency domain rolling causality analysis

Submitted to Economies

We would like to thank the Editors of the Economies for giving us the opportunity to revise our work. As suggested, we have carefully addressed all the comments of the reviewers. Our responses, indicated in the yellow color highlight, discuss and explain in detail the responses to the reviewers’ comments. Part or all of those responses has been added to the revised manuscript or new version of the paper.

Reviewer 5

Comments and Suggestions for Authors

This paper is analyzing an interesting topic.

 Reply: many thanks dear referee for being positive about the manuscript and making your comments and suggestion which we have considered with great care.

The following points should be clarified :

1. Justification on the sample chosen. Recommendation (NIFTY Mid Cap, NIFTY Small Cap, may be analyzed. if author/s find common result, so the finding will be more valuable and generalizable).

Reply: We are thankful for this valuable suggestion. But,  Nifty Small cap index is not traded in options market. Further, though NIFTY mid Cap index is available for trading in options market, it is very illiquid. Moreover, NIFTY 50 option market is taken because NIFTY50 represents 65% of the total market capitalisation of NSE.

2. Literature review section should be improved and the gap should be more clearly addressed.

Reply: We are thankful to the referee for this suggestion. Although literatures are few on put –call ratio, we have discussed following additional two papers on US market in the literature.

Blau, B. M., & Brough, T. J. (2015). Are put-call ratios a substitute for short sales?. Review of Derivatives Research18(1), 51-73.

Billingsley, R. S., & Chance, D. M. (1988). Put-call ratios and market timing effectiveness. Journal of Portfolio Management15(1), 25.

3. Plagiarism rate is quite high (20%).  

Reply: Thanks for this valuable comments. We are hopeful of reduction of plagiarism in the new and revised manuscript.

4. Conclusion should be enhanced, especially the part related to the findings. 

Reply: Thanks for the suggestion. We have further discussed the findings and enhanced the conclusion part as follows.

Extending the prior research relating to informational role of derivative market in general and option market in particular, the study examined the informational efficiency of volume and open interest PCR in predicting the market return and its implication for the traders and portfolio managers. First time we have studied the efficiency of the PCR at different frequencies and also the results are tested in an out of sample forecasting exercises in a rolling frequency domain causality framework. Firstly, the Granger causality from PCR to market return varies across the frequencies. Secondly, Long run causality is observed from open interest PCR to market return corresponding to time period of 12 days. In short run corresponding a time period of 2.5 days, volume PCR Granger causes market return. Thus traders and portfolio managers should use the appropriate PCR at the different time period in predicting a market return for trading and investment. In addition, unlike in long-run, in short-run volume PCR holds the predictability of market return during crisis period. Further, our findings are robust even after controlling for the information generated from futures market. The future study in this regard could be extended to effectiveness of PCR ratios across maturity and moneyness of the index options and also to stock options.  

Round 2

Reviewer 3 Report

Please read my first report and make additional revisions.